# Sequence specificity of an essential nuclear localization sequence in Mcm3

Ziyi Wang[1,2], Yun Jing Zhang[2], Qian-yi Zhang[1], Kate Bilsborrow[2], Matthew Leslie[2], Raymond T. Suhandynata[3], Huilin Zhou[1,2,4]*

1 Biomedical Science Graduate Program, University of California San Diego, San Diego, California, United States of America, 2 Department of Cellular and Molecular Medicine, University of California San Diego, San Diego, California, United States of America, 3 Department of Pathology, University of California San Diego, San Diego, California, United States of America, 4 Moores Cancer Center, School of Medicine, University of California at San Diego, San Diego, California, United States of America

* huzhou@health.ucsd.edu

## Abstract

Proteins with nuclear localization sequences (NLSs) are directed into the cell nucleus through interactions between the NLS and importin proteins. NLSs are generally short motifs rich in basic amino acids; however, identifying NLSs can be challenging due to the lack of a universally conserved sequence. In this study, we characterized the sequence specificity of an essential and conserved NLS in Mcm3, a subunit of the replicative DNA helicase. Through mutagenesis and AlphaFold 3 (AF3) modeling, we demonstrate that the precise positioning of basic residues within the NLS is critical for nuclear transport of Mcm3 through optimal interactions with importin. Disrupting these interactions impairs the nuclear import of Mcm3, resulting in defective chromatin loading of the MCM complex and poor cell growth. Our results provide a structure-guided framework for predicting and analyzing monopartite NLSs, which, despite lacking a single consensus sequence, retain key characteristics shared between the NLSs of Mcm3 and the SV40 large T antigen.

## Author summary

Transporting proteins into and out of the cell nucleus is essential for chromosome-associated activities. Nuclear localization sequences (NLSs), short motifs rich in basic amino acids, are commonly found in nuclear proteins. NLSs work by interacting with importin, a key transport receptor responsible for recognizing and guiding NLS-containing proteins through the nuclear pore complex into the nucleus. Other than being rich in basic amino acids, NLSs generally lack a discernible consensus sequence, raising questions about how they specifically control nuclear transport through their interactions with importins. Through a detailed mutagenesis study of a conserved and essential NLS in Mcm3, a subunit of the replicative DNA helicase, we demonstrate that the use of AlphaFold 3 (AF3), alongside genetic, biochemical, and cell biological analyses, define key contacts between Mcm3's NLS and importin that are required for nuclear import of Mcm3.

**Data availability statement:** The authors confirm that all data underlying the findings are fully available without restriction. All relevant data are within the paper and its Supporting information files.

**Funding:** This work was supported by the NIGMS (GM116897 to HZ) and by NIH (GM151191 to RTS and HZ). The funders had no role in study design, data collection and analysis, decision to publish, or preparation of the manuscript.

**Competing interests:** The authors have declared that no competing interests exist.

## Introduction

The nuclear localization sequence (NLS), a short motif rich in basic amino acids, controls the import of many nuclear proteins [1–13]. Two types of NLS have been identified: monopartite and bipartite [1–3,11–13]. A monopartite NLS consists of a single, continuous stretch of basic amino acids, including lysine and arginine [1–3,11,13]. The NLS from SV40's large T antigen ($^{126}$PKKKRKV$^{133}$), which contains five contiguous basic residues, is a classic example of a monopartite NLS [11]. In contrast, a bipartite NLS is characterized by two clusters of basic amino acids separated by a short spacer region, as seen in proteins such as nucleoplasmin and many others [1–3,12,13].

Both monopartite and bipartite NLSs function by interacting with proteins in the importin family, which are key transport receptors that recognize and guide NLS-containing proteins through the nuclear pore complex into the nucleus [1,14]. Structural studies have provided substantial insights into the interaction between NLSs and importin alpha, revealing that this binding involves both hydrophobic and charge-charge interactions, with multiple basic residues in the NLS contacting negatively charged residues on importin alpha [4,15–18]. This is supported through genetic studies, which illustrate that mutating Lys-128 in the SV40 NLS to non-basic residues significantly impaired nuclear import [11,19,20], whereas mutating it to arginine disrupts nuclear import without affecting viral replication [20]. In contrast, mutation of a nearby basic residue in the SV40 NLS (Lys-131) to methionine did not significantly affect nuclear import [20], suggesting that specific basic residues within the SV40 NLS may play distinct roles in controlling nuclear import.

In addition to the SV40 NLS, other monopartite NLSs have been identified, including those in H2B, v-Jun, and various others proteins [1–10,13]. Although these NLSs, like the SV40 NLS, are rich in basic residues, sequence variations among them raise questions about whether they interact with importin alpha in a similar manner [4,15]. The yeast *Saccharomyces cerevisiae* Mcm3, a subunit of the Mini-Chromosome Maintenance (MCM) complex [21–24], has a monopartite NLS ($^{764}$KSPKKRQRV$^{772}$) that resembles the SV40 NLS but has yet-to-be fully characterized [8,9]. In addition to the sequence similarity, several lines of evidence suggest that this sequence acts as a functional NLS for Mcm3. First, the SV40 NLS was shown to functionally substitute for Mcm3$^{757-779}$, providing strong support for the role of this Mcm3 fragment in controlling nuclear import [8,9]. Second, fusion of Mcm3$^{755-781}$ with galactosidase was sufficient to direct its nuclear localization [9]. Third, fluorescence microscopy of a GFP-tagged, shorter fragment of Mcm3$^{766-772}$ resulted in a modest nuclear localization signal in live cells [8], suggesting that the flanking sequences of this Mcm3 NLS are needed for optimal NLS activity. Despite these findings, the sequence specificity of Mcm3's NLS remains to be determined. Further characterization of Mcm3's NLS will not only help elucidate its cell cycle dependent nuclear import and subsequent loading, but also refine predictions of other putative NLSs that may exist in many nuclear proteins.

Since the nuclear import of Mcm3 and its subsequent chromatin loading are essential for cell growth [8,9], we performed mutagenesis studies to identify specific basic residues within Mcm3's NLS that are critical for its function. To understand the effects of these mutations, we used AlphaFold 3 (AF3) to model the binding between Mcm3's NLS and Kap60, the importin alpha in yeast [4,25].

## Results

### Specific basic residues in the conserved motif of Mcm3's NLS contribute to optimal cell growth

Prior studies identified that Mcm3$^{755-781}$ was sufficient to mediate its nuclear import [9]. Sequence alignment of this fragment across various species revealed that Mcm3$^{764-771}$, a shorter

and flexible region within Mcm3 that is not visible in the published MCM double hexamer structures, is well-conserved in fungi (Fig 1A) [23,25–30]. This conservation suggests that Mcm3$^{764-771}$ is likely crucial for nuclear import, though sequences flanking this conserved region may also be necessary for optimal NLS activity. To evaluate this, we examined the effects of various *mcm3* truncation mutants on cell growth via plasmid shuffling (Fig 1B). While truncation of the N-terminal sequences (Mcm3$^{742-763}$) and C-terminal sequences (Mcm3$^{781-885}$) flanking Mcm3$^{764-771}$ does not affect cell viability, truncation of the residues immediately adjacent to the conserved motif results in impaired cell growth, as observed in the mcm3$^{Δ771-780}$ mutant (Fig 1B). Thus, residues neighboring the conserved motif (Mcm3$^{764-771}$) are needed for optimal cell growth.

To fully characterize the sequence specificity of the Mcm3 NLS, we performed alanine scanning mutagenesis and found that K768, a highly conserved residue across fungi, mouse, and human (Fig 1A), plays a critical role in supporting cell viability. Mutation of Mcm3$^{K768A}$ severely impairs cell viability, while mutation of Mcm3$^{R769A}$ results in a partial growth defect (Fig 1C). Single alanine substitutions of surrounding residues within or near the Mcm3 NLS have no effect on cell growth (Figs 1C and S1A). Strikingly, Mcm3$^{K768A}$ occasionally accumulates colonies with enhanced growth (survivors) on 5-FOA plates, which are eliminated by deletion of the *RAD52* gene, suggesting that these survivors arise from Rad52-dependent recombination events (Fig 1C).

To remove the complementing *URA3* plasmid expressing wild-type (WT) Mcm3, *mcm3Δ* cells were treated with 5-fluoroorotic acid (5-FOA). This allowed the determination of the growth phenotypes of *mcm3* mutants expressed from a *LEU2*-containing plasmid in the same cell. Cells were plated on media where the complementing *URA3* plasmid was retained (SC-Leu) or lost (5-FOA). *mcm3* mutants with impaired cell growth were summarized in the lower panel of Fig 1C, with "++" indicating wild-type cell growth, "+" indicating partially impaired growth, and "−" indicating severe growth defects.

Besides the prominent role of K768, other basic residues in the Mcm3 NLS can have redundant roles for optimal cell growth. To test this idea, we generated mutants in which two basic residues in Mcm3$^{764-771}$ were mutated to alanine (Fig 2A). Although Mcm3$^{K764A}$, Mcm3$^{K767A}$, and Mcm3$^{R771A}$ single mutants do not affect cell growth (Fig 1C), combining any of them with Mcm3$^{R769A}$ severely impairs cell growth (Fig 2A). Like Mcm3$^{K768A}$, these growth-impaired double alanine mutants also accumulate Rad52-dependent survivors on 5-FOA plates (Fig 2A). In contrast, combining Mcm3$^{R769A}$ with Mcm3$^{R773A}$, a residue outside the conserved Mcm3 motif, does not affect cell growth (S1B Fig). Moreover, combining any two of the Mcm3$^{K764A}$, Mcm3$^{K767A}$, Mcm3$^{R771A}$, and Mcm3$^{R773A}$ mutations also does not result in appreciable growth defects (S1C Fig). Thus, basic residues within the Mcm3 NLS are not functionally equivalent: K768 is the most critical, R769 plays a secondary role, and K764, K767 and R771 also contribute to cell growth when R769 is mutated to alanine. This finding aligns well with the fact that K768 is the only invariant lysine residue in this region across evolution (Fig 1A). Within Mcm3's NLS, K767 and R769 flank K768, forming a KKR motif. To determine whether the precise order of these basic residues affect cell growth, we generated a combination of mutants in which they were substituted with either lysine or arginine (Fig 2B). Interestingly, mutants with the sequence KRK or RRK display poor growth and accumulate Rad52-dependent survivors (Fig 2B); this indicates that arginine cannot functionally replace K768 in Mcm3, regardless of whether the neighboring residues are lysine or arginine.

Additionally, at least one lysine is required in this region of Mcm3 to control nuclear import. While the SV40 NLS can functionally substitute Mcm3$^{766-772}$, replacing it with an all-arginine (allR) version of the SV40 NLS ($^{766}$PRRRRRV$^{772}$) fails to support cell growth and results in the accumulation of Rad52-dependent survivors (Fig 2C).

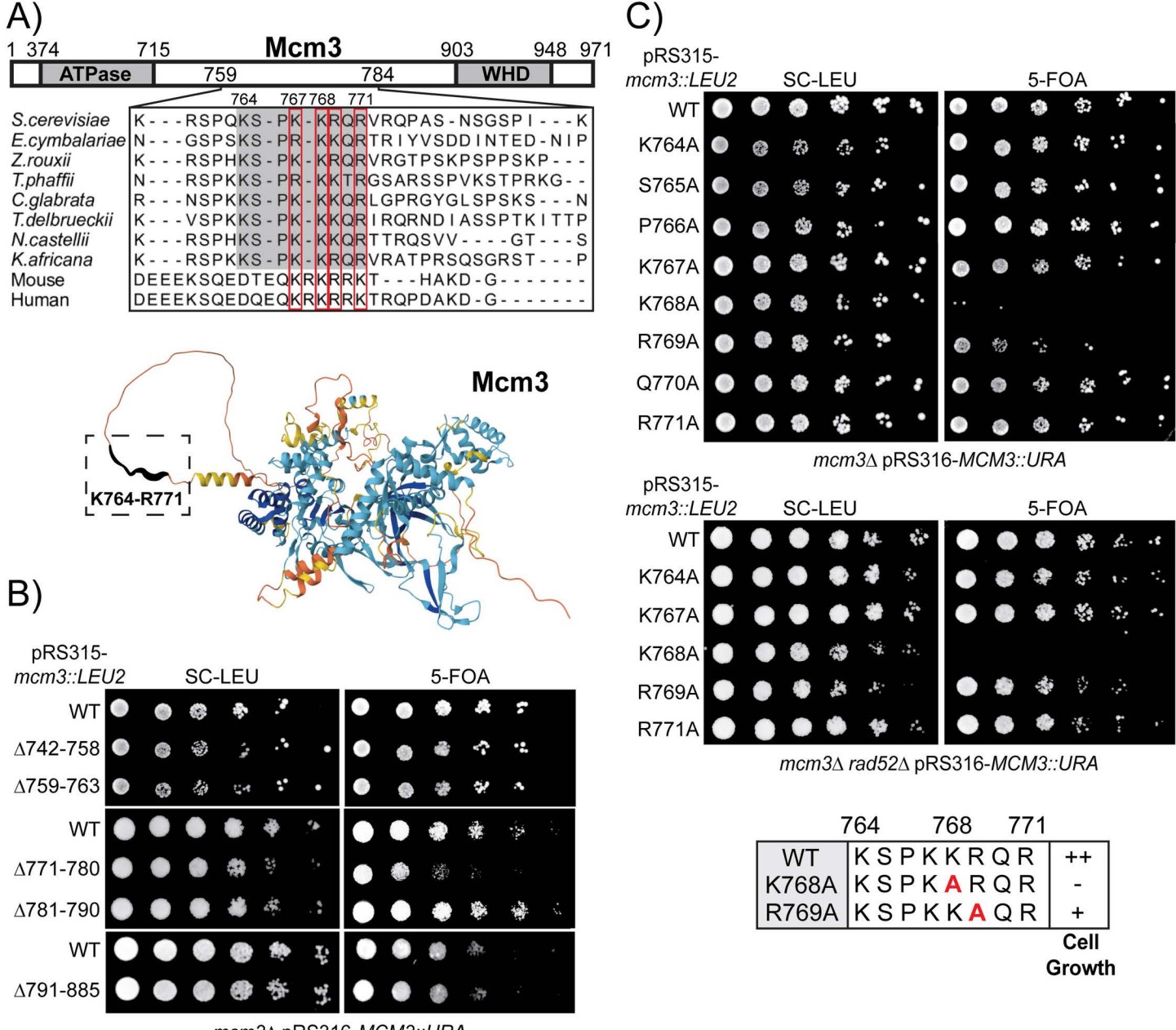

**Fig 1. Mutations within the conserved motif of Mcm3's NLS impair cell growth.** A) The conserved motif of Mcm3's NLS resides in an unstructured region. Upper panel: Sequence alignment illustrates conservation of the Mcm3 sequences from K764 to R771 in fungi, with K767, K768, R769 and R771 also conserved in mouse and human. Lower panel: AlphaFold predicts that the conserved motif (Mcm3[764-771], boxed with dotted lines) resides in a flexible region of Mcm3. B and C) *mcm3* mutants with the indicated mutations were analyzed by plasmid shuffling.

## Growth-impaired *mcm3* mutants fail to localize to the cell nucleus

Next, we used fluorescence microscopy to examine the localization of GFP-Mcm3 in cells, with Nup49-mCherry serving as a marker for the nuclear periphery. In agreement with previous studies [9,31–34], both wild-type (WT) GFP-Mcm3 and GFP-Mcm3 with the SV40 NLS substitution localize to the nucleus during G1 but not the G2/M phase (Fig 3A

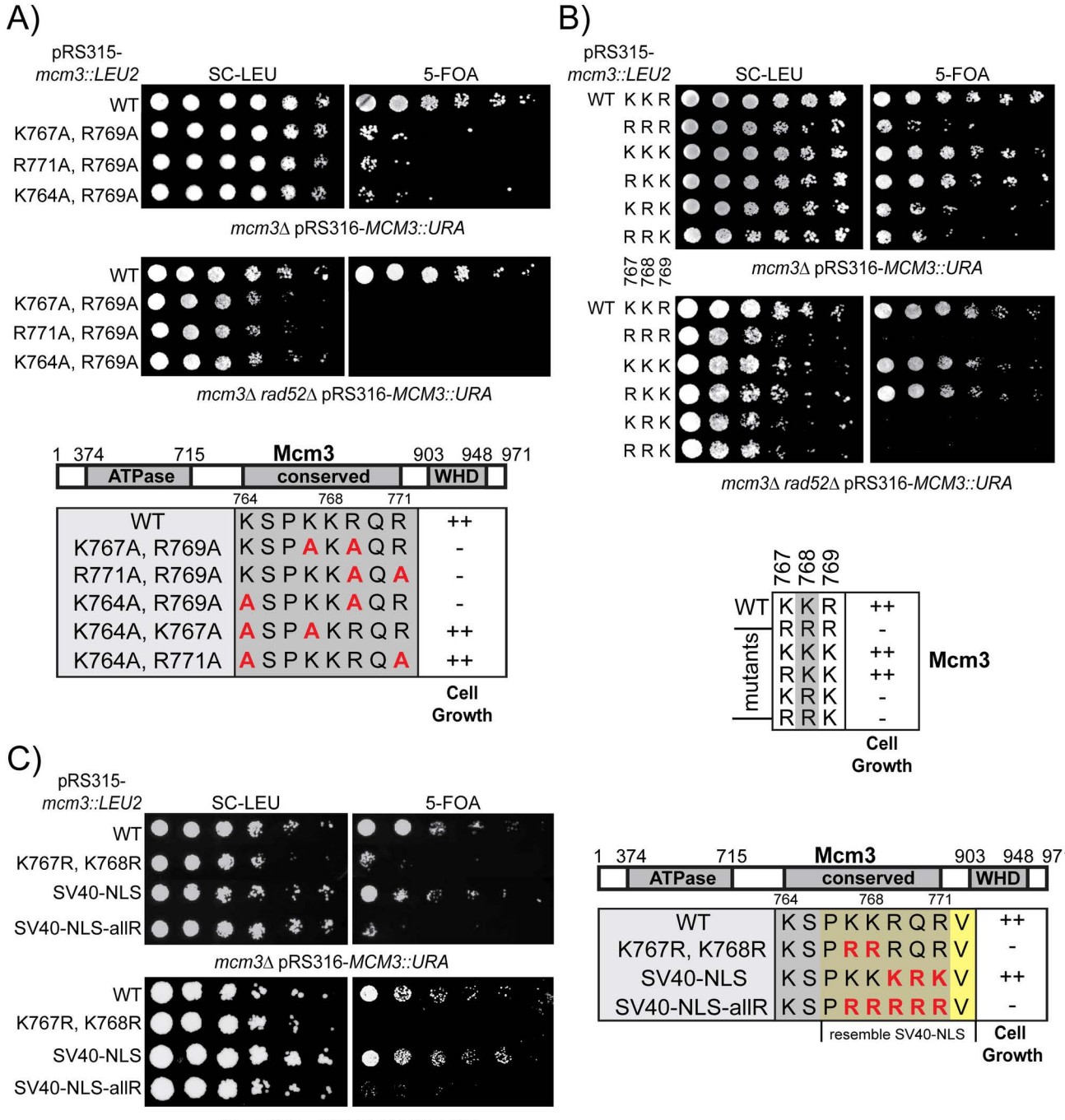

**Fig 2. Specific basic residues within the conserved motif of Mcm3's NLS are necessary for cell growth.** A, B and C) *mcm3* mutants with the indicated mutations were analyzed by plasmid shuffling. Mutants were plated on media where the complementing *URA3* plasmid was retained (SC-Leu) or lost (5-FOA). The effects of cell growth for mcm3 mutants were outlined in the lower panel of each figure, with "++" indicating wild-type cell growth and "−" indicating severe growth defects.

and [3B]). In contrast, Mcm3[K767R, K768R], Mcm3[SV40-NLS-allR] and other growth-impaired mutants with mutations in Mcm3[764-771] do not localize to nucleus in both G1 and G2/M phases (Figs 3 and S2).

A)

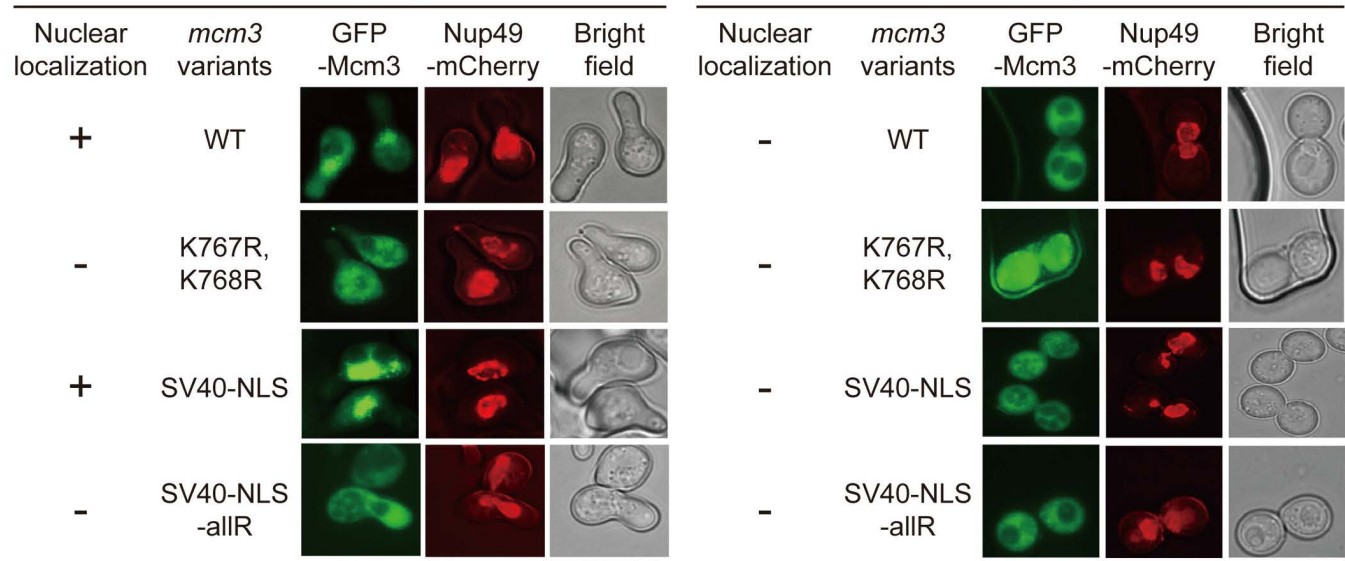

B)

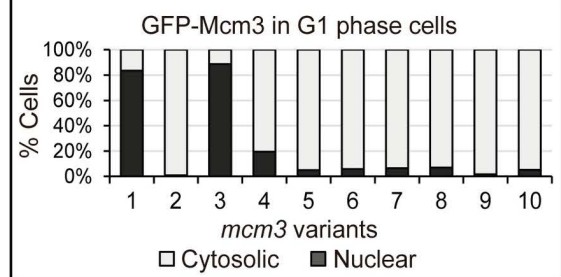
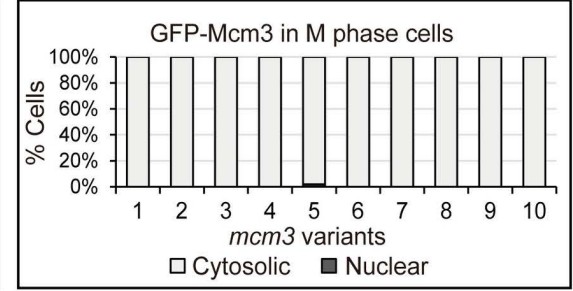

**Fig 3. mcm3 mutants with impaired cell growth do not localize to the nucleus.** A) Nup49-mCherry cells (HZY1575) transformed with pRS315-*GFP-mcm3* plasmids were arrested in G1 or M phase for fluorescence microscopy analysis. Representative images of cells in G1 (left panel) and M (right panel) phases illustrate the localization of GFP-Mcm3 relative to Nup49-mCherry. B) The percentage of cells with nuclear or cytosolic localization of GFP-Mcm3 is illustrated for *mcm3* variants, with representative images shown in Figs 3A and S2A. Detailed quantification data is provided in S2B Fig.

Since the fluorescent signal of GFP-Mcm3 cannot distinguish between chromatin-bound and free-floating Mcm3 in the nucleoplasm [23,35–38], we used a C-terminal fragment GFP-Mcm3$^{721\text{-end}}$, which cannot be loaded onto chromosomes, for fluorescence microscopy. Like GFP-Mcm3 (Fig 3), both C-terminal fragments (WT and SV40 NLS substitution constructs) localize to the cell nucleus in the G1 phase but not in the G2/M phase (Fig 4A and 4B). Notably, the nuclear signal of the Mcm3 fragment with the SV40 NLS substitution appears to have greater intensity compared to that of the native Mcm3 fragment, which has a modest yet detectable nuclear signal (Fig 4A and 4B). In contrast, all growth-impaired GFP-Mcm3 mutant fragments with mutations in Mcm3$^{764\text{-}771}$ do not localize to the nucleus in both G1 and M phases (Figs 4B and S3). Together, these findings suggest that mutations within the conserved motif of Mcm3's NLS disrupt its nuclear localization, leading to the observed growth defects of these mutants.

A)

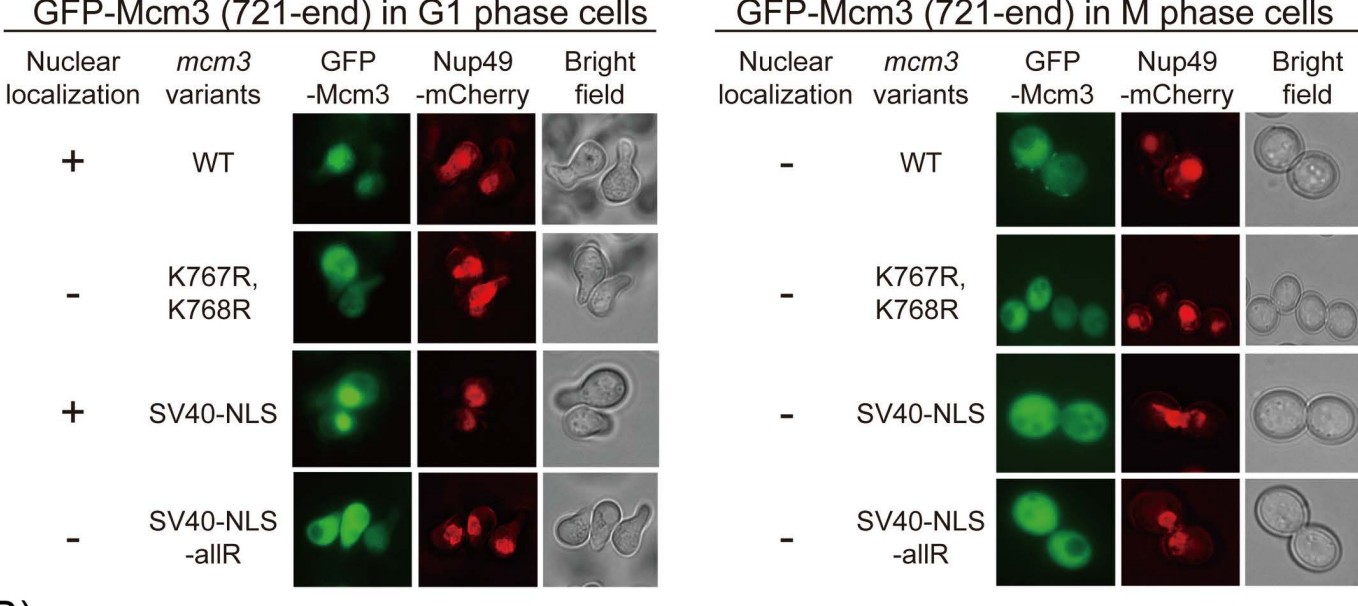

B)

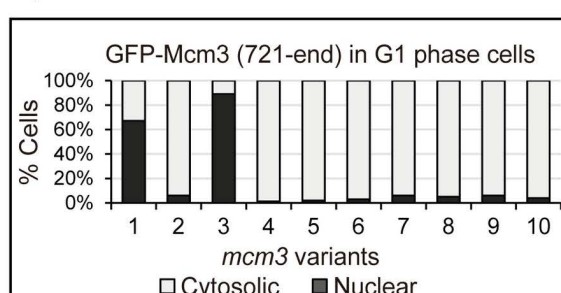

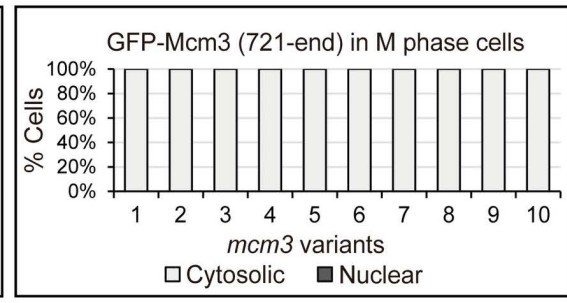

| | *mcm3* variants |
|---|---|
| 1 | WT |
| 2 | K767R, K768R |
| 3 | SV40-NLS |
| 4 | SV40-NLS-allR |
| 5 | K768A |
| 6 | K764A, R769A |
| 7 | K767A, R769A |
| 8 | R771A, R769A |
| 9 | K768R, R769K (KRK) |
| 10 | K767R, K768R, R769K (RRK) |

**Fig 4. The 721-end fragment of *mcm3* mutants with impaired cell growth do not localize to the nucleus.** A) Nup49-mCherry cells (HZY1575) transformed with pRS315-*GFP-mcm3-721-end* plasmids were arrested in G1 or M phase for fluorescence microscopy analysis. Representative images of cells in G1 (left panel) and M (right panel) phases illustrate the localization of GFP-Mcm3 (721-end) relative to Nup49-mCherry. B) Percentage of cells with nuclear or cytosolic localization of GFP-Mcm3 (721-end) is illustrated for *mcm3* variants, with representative images shown in Figs 4A and S3A. Detailed quantification data is provided in S3B Fig.

As previously reported, the absence of Mcm3 in the cell nucleus during the G2/M phase could be attributed to a leucine-rich nuclear export signal (NES) motif (Mcm3$^{834-842}$) [8]. This is supported by the constitutive nuclear localization signal observed in both full-length and C-terminal fragments of Mcm3 lacking this NES (Mcm3$^{\Delta781-884}$) (S4A and S4B Fig).

### Mcm3's NLS mutants affect chromatin association *in vivo* but not MCM loading *in vitro*

Defective import of Mcm3 into the cell nucleus may compromise MCM loading in cells, leading to severe growth defects (Figs 1 and 2). To test this hypothesis, we examined the chromatin association of MCM in cells arrested in the G1 or the G2/M phase (Fig 5A) [39,40]. We found that MCM subunits were specifically enriched in the chromatin fraction

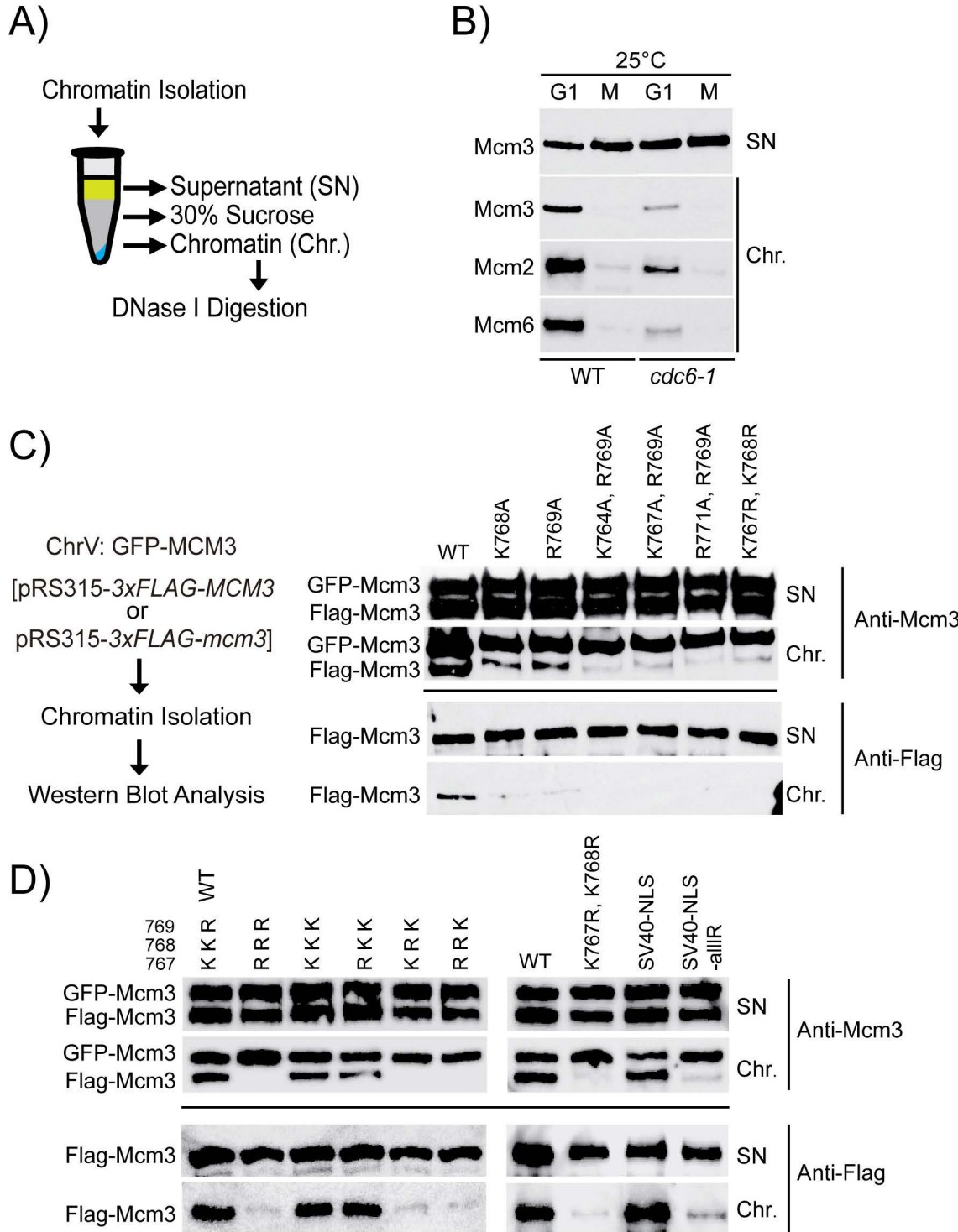

**Fig 5. mcm3 mutants with impaired cell growth exhibit chromatin association defects in the G1 phase.** A) The workflow of yeast chromatin isolation via centrifugation through a sucrose step. B) Wild-type (HZY4011) and *cdc6-1* (HZY1366) cells were arrested in the G1 and M phase at 25 °C. The isolated supernatant and chromatin fractions were analyzed by Western blotting with anti-Mcm3, anti-Mcm2, and anti-Mcm6 antibodies. C and D) GFP-Mcm3 cells (HZY3037) transformed with pRS315-*3xFLAG-mcm3* mutant plasmids were arrested in the G1 phase. The isolated supernatant and chromatin fractions were analyzed by Western blotting with anti-Mcm3 (for detecting both GFP-Mcm3 and Flag-Mcm3, upper panel) and anti-Flag (for detecting Flag-Mcm3 only, lower panel) antibodies.

of cell arrested in the G1 phase but not the G2/M phase (Fig 5B, with S5A Fig confirming cell cycle arrest). This cell-cycle dependency suggests that the chromatin-bound MCM likely represents loaded MCM complexes. This is further supported by the finding that the amount of chromatin-bound MCM in G1 cells is reduced by the partial inactivation of Cdc6 (Fig 5B, with S5A Fig confirming cell cycle arrest), validating that this approach specifically detects loaded MCM complexes.

Next, we utilized this assay to evaluate MCM-chromatin association in the growth-impaired *mcm3* mutants described in Figs 1 and 2 (Fig 5C and 5D, with S5B–D Fig confirming cell cycle arrest). In each sample, wild-type GFP-Mcm3 displays robust chromatin association among the cells analyzed. By comparison, all growth-impaired *mcm3* mutants exhibit a significantly reduced amount of chromatin-bound Mcm3.

We considered that these *mcm3* mutants could affect the formation of MCM complex, leading to the observed defective chromatin loading. To test this, we examined the ability of these *mcm3* mutants to be loaded on DNA *in vitro*. To do so, we performed an *in vitro* MCM loading assay using whole cell extract (WCE) supplemented with purified Cdc6 and ORC proteins [41]. We observed that mutations within the conserved motif of Mcm3's NLS do not affect the loading of MCM complexes onto the ARS-containing DNA substrate *in vitro* (S6 Fig). Therefore, the observed reduction in MCM-chromatin association is not because these *mcm3* mutants cannot be loaded as the MCM complex, but rather it is because they are not imported into the nucleus.

**A structure-guided approach to understand the specificity of Mcm3's NLS.** To evaluate how Mcm3's NLS interacts with importin alpha, we applied AF3 to model the interactions between Mcm3's NLS ($^{764}$KSPKKRQRV$^{772}$), or the SV40-NLS substitution ($^{764}$KSPKKKRKV$^{772}$), and the importin alpha of *Saccharomyces cerevisiae* (Kap60) [42]. We found that K128 at Position 2 (P2) of the SV40 NLS (KKKRK) and K768 of Mcm3 (KKRQR) contact the same three residues in the major binding pocket of importin alpha (Gly161, Thr166, and Asp203) at distances of 2.8 Å to Thr166 and 2.6 Å to Asp203 (Fig 6A), similar to contacts observed in the X-ray structure of the SV40 NLS-importin alpha complex [4]. Moreover, R769 of Mcm3 contacts the same residues (Glu272 and Asp276) of Kap60 in the major binding pocket as K129 of the SV40 NLS at Position 3 (P3) (KKKRK) (Fig 6B). Thus, the conserved interactions at P2 and P3 appear to be critical for the binding between Mcm3's NLS and Kap60; these interactions may explain the important role of K768 and R769 in supporting cell growth via nuclear import of Mcm3.

Next, we utilized AF3 to model the interaction between Kap60 and various mutant forms of Mcm3's NLS, focusing on the interactions at P2 and P3 positions (Figs 6C and S7). For mutants with no significant growth defects, the interactions between the basic residues at P2 or P3 and Kap60 resemble those in wild-type Mcm3. Mutations at P1 and P5 (e.g., Mcm3$^{K767A}$ and Mcm3$^{R771A}$) do not disrupt the interactions between Kap60 and residues at P2 or P3 within Mcm3's NLS. All growth-impaired mutants, except for Mcm3$^{SV40-NLS-allR}$, exhibit compromised/altered interactions at P2 or P3 with Kap60. AF3 cannot accurately predict the extent of defects among various growth-impaired *mcm3* mutants. For instance, like the other *mcm3* mutants with severe growth defects, AF3 modeling indicates that Mcm3$^{R769A}$ mutants have altered interactions with Kap60 at P2 and P3; however, it exhibits only partial growth impairment (Fig 1C) [41].

## Discussion

The classical model of nuclear localization sequences (NLSs) suggests that clusters of basic residues—lysine and arginine—form either monopartite or bipartite NLSs [1–3,11–13]. Thus far, no single consensus sequence for NLSs has been definitively identified. The lack of consensus

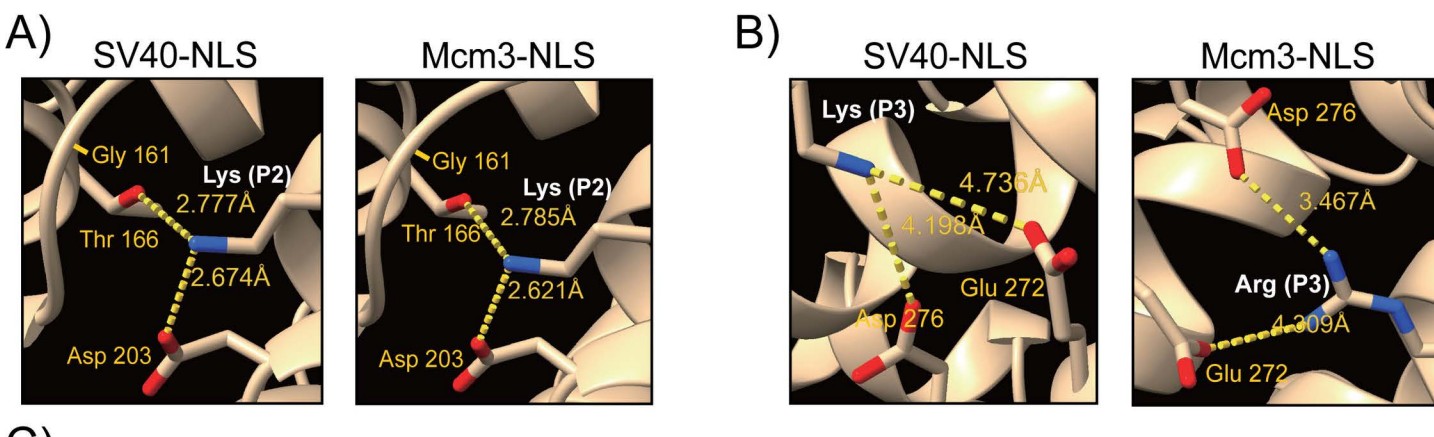

**Fig 6. AlphaFold predicted interactions of mutant Mcm3's NLS with Kap60.** A) AlphaFold predicted interaction of Kap60 and the lysine at position 2 (P2, colored in white) of Mcm3's NLS ($^{764}$KSPKKRQRV$^{772}$), or the SV40-NLS substitution construct ($^{764}$KSPKKKRKV$^{772}$). B) AlphaFold predicted interaction of Kap60 and the basic residue at position 3 (P3, colored in white) of Mcm3's NLS ($^{764}$KSPKKRQRV$^{772}$), or the SV40-NLS substitution construct ($^{764}$KSPKKKRKV$^{772}$). C) The interaction between Kap60 and the residues at P2 or P3 within the mutant Mcm3's NLS was modeled using AlphaFold 3, and then compared to the published structure of the SV40-NLS/karyopherin alpha complex. Green indicates the *mcm3* mutant (Mcm3$^{SV40-NLS-allR}$), where the mutagenesis results are not in agreement with AlphaFold predictions. Orange indicates mutations that severely impaired nuclear localization and cell growth. Residues in red are essential for optimal NLS function. Grey indicates contacts between residues at P2 or P3 within the Mcm3's NLS and Kap60 that deviate from the interactions observed in the published SV40-NLS/karyopherin alpha complex.

sequence reflects the flexible interactions between different NLSs and importin alpha, each with varying specificities and capabilities in recognizing and transporting different NLS-containing cargo proteins [3,15,16,19,43]. As a result, it is difficult to predict an NLS based on its amino acid sequence alone. Here, we used mutagenesis to investigate the sequence specificity of Mcm3's essential NLS and utilized AF3 to model its interaction with importin alpha.

Comparison between these results and the X-ray structure of the SV40 NLS-importin alpha complex suggests that specific contacts between importin alpha and Mcm3's NLS at P2 and P3 are crucial for nuclear import [4]. These findings suggest that AF3 modeling can be used to evaluate monopartite NLSs, which may lack a single consensus sequence but still retain the key characteristics shared with SV40-NLS and Mcm3-NLS to stimulate further experimental studies.

### Defining nuclear localization sequences (NLSs) through AF3

AF3 is the latest tool capable of predicting protein-protein interactions [43]. Here, we utilized this tool to model the interaction between importin alpha and Mcm3's NLS variants. Our analysis demonstrates that the NLSs of Mcm3 and SV40 share similar contacts with Kap60 (Figs 6A, 6B and S7B). Specifically, K768 of Mcm3 and K128 of SV40 occupy P2 [4], while R769 of Mcm3 and K129 of SV40 occupy P3 with regard to their contacts with Kap60; these basic residues are needed for optimal cell growth (Figs 1 and 2). Additionally, basic residues at P1 and P5 play a supporting role for optimal nuclear localization, likely by contributing to the binding between Kap60 and P2 or P3 of the NLS [4,15,16,18]. In contrast, P4 is often occupied by a non-basic residue in monopartite NLSs [5,7]; structural studies of the SV40 and c-Myc NLSs suggest that P4 does not interact specifically with importin alpha, thus P4 may be less critical for nuclear import [4,15]. In support of the essential role of P2 and P3 mediated interactions, mutations of Mcm3 at P2 or P3 result in defective nuclear import, reduced chromatin association, and impaired cell growth. These findings underscore the structural and functional parallels between the NLS of Mcm3 and SV40, highlighting the central role of P2 and P3 mediated interactions with Kap60.

Previously, the K/R-K-X-K/R motif (where X can be any residue) was proposed as a possible consensus for a monopartite NLS [1,4,44]. However, our findings suggest that this consensus model may be overly simplistic. For example, variants of Mcm3's NLS such as Mcm3$^{R771A}$ ($^{764}$KSPKKRQ$\underline{A}$V$^{772}$) can still localize to the nucleus, as indicated by their normal cell growth (Fig 1). Similarly, other NLSs, such as that from V-Jun (KSRKRVL), also deviate from this proposed consensus sequence [1,6,45]. Instead, we propose that AF3 offers a new approach to evaluate monopartite NLSs.

Insights gained from this study, along with previously published, well-characterized NLSs [4–7,11,16], suggest several key considerations for analyzing monopartite NLSs. First, a basic residue (more commonly a lysine) at P2 plays the most critical role in recognizing importin alpha. Second, the basic residue at P2 must be surrounded by at least two basic residues, which can be located at P1, P3, P5, or, in some cases, P4. In rare cases, even more distant residues can contribute to NLS function, such as K764 of Mcm3, which is at the minus 3 position relative to P1. These features, along with the use of AF3, can be used to study putative NLSs.

### Future directions

Although AF3 modeling predicts altered contacts between the NLSs of growth impaired *mcm3* mutants and Kap60, predicting the severity of growth impairment remains challenging. For example, similar to other *mcm3* mutants with severe growth defects, the contacts mediated by basic residues at P2 and P3 of Mcm3$^{K768R}$ and Mcm3$^{R769A}$ deviate from those of wild-type Mcm3 (Figs 6C and S7B), yet they only show partial growth defects (Fig 1C) [41]. Thus, further biophysical studies, including measurements of binding affinity between Mcm3's NLS and Kap60, are needed to characterize various *mcm3* mutations and gain a deeper understanding of their effects on nuclear import.

An unexpected finding from our study is that the Mcm3$^{SV40-NLS-allR}$ mutant fails to support nuclear import of Mcm3 (Figs 3 and 4), despite AF3 predicting similar contacts with Kap60 at

P2 and P3 (S7 Fig). This could be due to the inability of AF3 in distinguishing between basic residues (e.g., the primary amine of lysine versus the guanidine group of arginine). Moreover, we have previously shown that K767 and K768 in Mcm3 are redundantly modified by small ubiquitin-like modifier (SUMO) [41], a post-translational modification that cannot occur on arginine. This study provides additional support of the importance of K767 and K768 in Mcm3, which are critical for both its nuclear import through interaction with Kap60 and loading onto chromatin through modification by SUMO [41].

The defects of growth-impaired *mcm3* mutants in nuclear import and chromatin association are considered a loss-of-function. However, the accumulation of Rad52-dependent survivors during plasmid shuffling is not explained solely by a loss-of-function defect, as wild-type Mcm3 is present prior to its removal by plasmid shuffling (Figs 1 and 2). We have previously shown that these survivors are a product of gene conversion between the wild-type Mcm3 and the *mcm3* mutant [41]. The frequency of these survivors was low and variable, and their improved growth made it impractical to measure using the classical fluctuation analysis. The existence of Rad52-dependent survivors suggests that an unknown gain-of-function effect caused by these *mcm3* mutants may have induced double-stranded DNA breaks in the *mcm3* plasmids. Thus, further studies are needed to understand this defect, which may reveal additional functions of Mcm3's NLS beyond its role in nuclear import.

## Materials and methods

### Yeast genetics method and plasmid construction

Standard yeast genetics methods were used to construct the strains used in this study (S1 Table). Unless noted otherwise, plasmids were constructed using DNA recombination repair in yeast via >35 base pairs of homologies among PCR fragments used. Plasmids were rescued from yeast via electroporation, purified from bacteria and confirmed by DNA sequencing (S2 Table). Details of these constructions are available upon request.

### Spotting assay

Yeast cells were grown in 5 mL of SC-LEU media at 30 °C until saturation. Cultures were normalized to an $OD_{600}$ of 1.0, followed by 5-fold serial dilutions using multichannel pipettes in a 96-well plate. From each dilution, 8 μL of cells were spotted on SC-LEU or 5-FOA media. Plates were incubated at 30 °C for 3 days, and then imaged using a Bio-Rad ChemiDoc imaging system.

### Fluorescence microscopy

Yeast cells were synchronized in the G1 phase or G2/M phase by treatment with 15 nM alpha factor or 20 μg/mL Nocodazole in 5 mL of SC-LEU media for 4 hours. Live Cells were then immobilized on glass slides coated with 0.2 μg/μL Concanavalin A to prevent cell movements during imaging. Imaging was performed using the ECHO Revolve microscope equipped with an Olympus Plan Apo 60x oil objective lens (NA 1.42, WD 0.15 mm), 5 MP CMOS monochrome camera for fluorescence and 12 MP color camera for brightfield. Fluorescence detection was conducted using appropriate filter sets: the FITC filter for GFP detection and the Texas Red filter for mCherry fluorescence. Exposure times were standardized across all samples to ensure comparability within each experiment, with 410 ms for GFP (FITC filter) and 1125 ms for mCherry (Texas Red filter). All images were captured under identical conditions to minimize variation between samples. For publication purposes, all representative images were processed using Adobe Photoshop 2024, where brightness and contrast adjustments were made to enhance image clarity and presentation, while maintaining the integrity of the data.

## Chromatin isolation

Yeast cells were cultured in SC-LEU media (for *mcm3* mutants) or YP (for wild-type and *cdc6-1* cells) containing 2% glucose and grown overnight to $OD_{600nm}$ of 0.3. Cells were arrested with 15 nM alpha-factor at the G1 phase or 20 μg/mL Nocodazole at the G2 phase for 4 hours. For each sample, approximately 50 ODml cells were collected, washed, and stored in 20% glycerol at −20 °C before use. Cell pellets were resuspended in 1 mL of buffer (100 mM PIPES pH 8.5, and 10 mM DTT) and incubated at 30 °C for 15 minutes. After incubation, 1 mL of spheroplasting solution (50 mM $NaPO_4$ pH 7.5, 1.0 M sorbitol) and 300 μL of 0.5 mg/mL Lyticase were added to each cell pellet, and the mixture was incubated at 30 °C for 1 hour to remove the cell wall. Spheroplasts were pelleted down at 4 °C and washed once with 1 mL of ice-cold wash buffer (0.1 M NaCl, 50 mM HEPES-KOH pH 7.5, 1.0 M sorbitol) to remove residual Lyticase. For cell lysis, spheroplast pellets were resuspended in 1 mL of lysis buffer (50 mM HEPES-KOH pH 7.5, 100 mM NaCl, 10 mM MgOAc, 0.5% Triton X-100, 1 mM EDTA, protease inhibitors) and rotated at 4 °C for 10 minutes. Half of the lysate was then layered on top of an equal volume of lysis buffer containing 30% sucrose and centrifuged at 15,000 rpm for 15 minutes at 4 °C. The supernatant was collected for further analysis. The pellet was washed once with 1 mL of lysis buffer at 4 °C for 30 minutes. To solubilize chromosome-bound materials, crude chromatin pellets were digested in 100 μL of DNase I buffer (50 mM HEPES-KOH pH 7.5, 100 mM NaCl, 50 mM $MgCl_2$, 10 mM $CaCl_2$, 0.1% NP-40, 1 mM EDTA, protease inhibitors) with 5 μL of Turbo DNase I (Invitrogen) at 4 °C overnight. The next day, the mixture was centrifuged, and the supernatant portion (DNase I elution) was recovered for further analysis. For Western blot, DNase I elution samples were normalized based on the concentration of supernatant (cytosolic materials) after sucrose step centrifugation.

## Lyticase purification

Lyticase was cloned and purified as previously described [46] with modifications. Briefly, Rosetta cells expressing lyticase were cultured in 6-liters LB medium supplemented with ampicillin and grown at 37 °C overnight until saturation. The culture was spun down, and the cell pellets were washed once with 500 mL of Tris buffer (25 mM Tris pH 7.4). The cell pellets were then resuspended in 125 mL of buffer (25 mM Tris pH 7.4, 2 mM EDTA), mixed with an equal volume of Tris buffer (25 mM Tris pH 7.4) containing 40% sucrose, and stirred gently at room temperature for 20 minutes. Following centrifugation, the supernatant was discarded. The cell pellets were gently resuspended in 125 mL of cold 0.5 mM $MgSO_4$ and stirred for 20 minutes at 4 °C. Upon centrifugation, the supernatant containing lyticase was collected and stored in 10 mM Tris pH 7.4 at −80 °C.

## Preparation of cell lysates for MCM loading assay

Yeast cells were cultured in 200 mL of SC-LEU media containing 2% glucose and grown overnight to saturation. The following day, 100 mL of saturated yeast culture was diluted into 1-liter YPD media and refreshed for 4 hours. Subsequently, the cells were arrested in the G1 phase using 20 nM alpha-factor for 4 hours. After the arrest, cells were harvested by centrifugation, washed once with Buffer L (25 mM HEPES-KOH pH 7.8, 2 mM $MgCl_2$, 0.1 mM EDTA, 0.5 mM EGTA, 0.1% NP-40, 175 mM K-Glutamate, 15% glycerol, protease inhibitors), and resuspended in a volume equivalent to one-quarter pellet size of Buffer L. The cell suspensions were then flash-frozen in liquid nitrogen as droplets and stored at −80 °C. Whole-cell lysates were prepared by mechanically grinding the frozen droplets into fine powders using FreezerMill (SPEX SamplePrep). To extract chromosome-bound ORC and MCM complex, 3M K-glutamate was added to the thawed powder to a final concentration of 300 mM and incubated

at 4 °C for 30 minutes. The lysates were then centrifuged, and the clear supernatant was collected and stored at −80 °C. Typical concentration of lysates was determined to be 60 mg/mL by Bradford assay. For the MCM loading assay, lysates were diluted by Buffer L with no K-glutamate to reach the desired protein concentration.

## Preparation of ARS209-containing DNA beads for MCM loading assay

A 700 bp ARS209-containing DNA fragment was PCR amplified from a plasmid (HZE3047) using 5' biotinylated primers (5Biosg/CTGCTCTGATGCCGCATAG, 5Biosg/CAGGAAG-GCAAAATGCCGC). The PCR product was precipitated by 70% ethanol and washed once with a mixture of 70% ethanol/ 30% 0.1M ammonium acetate to remove primers. The biotinylated ARS209 fragment was bound to Streptavidin M-280 Dynabeads (Invitrogen) and then incubated with free Streptavidin (Promega) to block free DNA ends. The typical capacity of Streptavidin beads for binding to ARS DNA is 3–5 μg DNA per 100 μL beads.

## ORC and Cdc6 purification

Cdc6 was purified as previously described [41,47,48] with minor modifications. Briefly, Rosetta cells expressing GST-Cdc6 (HZE1134) were cultured in 1-liter LB medium supplemented with ampicillin and chloramphenicol. The culture was grown at 37 °C overnight until reaching an $OD_{600nm}$ of 0.5. The next day, cells were cooled on ice for 5 minutes, supplemented with 0.2 mM IPTG, and then induced for expression at 17 °C overnight. The following day, cells were harvested by centrifugation, washed once with 50 mL PBSN (1.06 mM $KH_2PO_4$, 5.6 mM $K_2HPO_4$, 154 mM NaCl, 0.2% NP-40, and protease inhibitors), resuspended in 10 mL PBSN, and lysed by sonication. Upon centrifugation, the supernatant was collected and incubated with 500 μL of packed Glutathione beads (GE Healthcare) at 4°C for 2 hours. After incubation, the beads were washed with 30 mL of PBSN followed by 20 mL of Buffer L at 4°C. GST-Cdc6 was eluted using 2 mL of Buffer L supplemented with 10 mM glutathione and 20 mM NaOH.

ORC was purified as previously described [26,41] with minor modifications. Briefly, yeast cells (HZY1785) were grown in 1-liter YP-Raffinose to an $OD_{600}$ = 0.5. ORC expression was induced with 1.5% galactose overnight. The following day, saturated cells were harvested by centrifugation, washed with Buffer C (25 mM HEPES-KOH pH 7.8, 0.05% NP-40, 0.1 M KCl, 10% glycerol), and resuspended in a volume equivalent to one-half pellet size of Buffer C. The cell suspensions were then flash-frozen in liquid nitrogen as droplets and stored at -80°C. Whole-cell lysates were prepared by mechanically grinding the frozen droplets into fine powders using FreezerMill (SPEX SamplePrep). To extract chromosome-bound ORC, 4M KCl was added to the thawed powder to a final concentration of 300 mM and incubated at 4 °C for 30 minutes. The thawed powder was then centrifuged, and the clear supernatant was collected. Subsequently, 500 mM $CaCl_2$ was added to the clear lysate to achieve a final concentration of 5 mM. The lysates were then bound to 2.7 mL packed Calmodulin beads (GE Healthcare) and incubated at 4°C for 2.5 hours. After incubation, the beads were washed with 120 mL ice-cold wash buffer (300 mM KCl, 5 mM $CaCl_2$, 1 mM DTT). ORC was eluted with 7.5 mL ice-cold elution buffer (300 mM KCl, 10 mM EGTA, 5 mM EDTA, 20% glycerol, and protease inhibitors).

## *In vitro* MCM loading assay

For each MCM-DH loading reaction, 100 μL of 30 mg/mL G1 cell extracts, 2 μL of 1 mg/mL salmon sperm DNA, 235 nM Cdc6, and 94 nM ORC were incubated with Streptavidin beads containing 1.5 pmol biotinylated DNA with or without 20X ATP regeneration system (0.1 M

Tris-HCl pH 7.5, 20 mM $MgCl_2$, 15% glycerol, 350 unit/mL Creatine kinase, 0.7 M phospho-creatine, 40 mM ATP). The reactions were performed at 25 °C for 30 min with rotation. After incubation, the unbound fractions were removed, and the beads were washed 4 times with 1 mL of ice-cold Buffer L. To elute DNA-bound materials, the beads were boiled with 50 μL 1X LDS containing 10 mM DTT for 5 min. The LDS elution was saved for western blotting analysis.

### Flow cytometry

Samples were prepared and analyzed using a BD LSR II Flow Cytometer as previously described [41]. Results were processed using FlowJo v10.6 for publication purposes.

### AlphaFold 3 (AF3) modeling of Mcm3's NLS/Kap60 interaction

The NLS sequence (amino acids 764-772) of *mcm3* variants and full-length Kap60 were used as input for AF3 modeling. The predictions with the highest confidence scores were selected for analysis. Structural outputs were analyzed using ChimeraX to identify the contacts and distances between the basic residues at P2 or P3 in the Mcm3 NLS and the corresponding residues in Kap60. These contacts were then compared to those found in the experimental structures of NLS-importin [4].

### Supporting information

**S1 Table.  Yeast cells used in this study.**
(EXCEL)

**S2 Table.  Plasmids used in this study.**
(EXCEL)

**S1 Fig.**  A, B and C) mcm3 mutants with the indicated mutations were analyzed by plasmid shuffling. Mutants were plated on media where the complementing *URA3* plasmid was retained (SC-Leu) or lost (5-FOA).
(TIF)

**S2 Fig.**  A) Nup49-mCherry cells (HZY1575) transformed with pRS315-GFP-mcm3 plasmids were arrested in G1 or M phase for fluorescence microscopy analysis. Representative images of cells in G1 (left panel) and M (right panel) phases illustrate the localization of GFP-Mcm3 relative to Nup49-mCherry. B) Quantification data for Fig 3B. The number of cells exhibiting either nuclear or cytosolic localization of the indicated GFP-Mcm3 variants, along with the total number of cells counted are indicated in the table.
(TIF)

**S3 Fig.**  A) Nup49-mCherry cells (HZY1575) transformed with pRS315-GFP-mcm3-721-end plasmids were arrested in G1 or M phase for fluorescence microscopy analysis. Representative images of cells in G1 (left panel) and M (right panel) phases illustrate the localization of GFP-Mcm3 (721-end) relative to Nup49-mCherry. B) Quantification data for Fig 4B. The number of cells exhibiting either nuclear or cytosolic localization of the indicated GFP-Mcm3 (721-end) variants, along with the total number of cells counted are indicated in the table.
(TIF)

**S4 Fig.**  A and B) Nup49-mCherry cells (HZY1575) transformed with pRS315-GFP-mcm3 or pRS315-GFP-mcm3-721-end plasmids with the indicated truncations were arrested in G1 or

M phase for fluorescence microscopy analysis. Representative images of cells in G1 (left panel) and M (right panel) phases illustrate the localization of GFP-Mcm3 (S4A Fig) or GFP-Mcm3 (721-end) (S4B Fig) relative to Nup49-mCherry.
(TIF)

**S5 Fig.** A) Fluorescence-Activated Cell Sorting (FACS) analysis of wild-type (HZY4011) and cdc6-1 (HZY1366) cells following G1 and M phase arrest for Figs 5B and S6C. B) FACS analysis of *mcm3* mutants following G1 arrest for Figs 5C, S6D to S6F. C) FACS analysis of *mcm3* mutants following G1 arrest for Figs 5D and S6G. D) FACS analysis of *mcm3* mutants following G1 arrest for Figs 5D and S6H.
(TIF)

**S6 Fig.** A) A workflow for loading MCM on ARS-containing DNA in vitro. B) Purified ORC and Cdc6 proteins were analyzed by SDS–PAGE and stained with Coomassie blue dye. * Indicates partial degradation of the proteins. C) G1 phase whole cell extracts (WCE) of wild-type (HZY4011) cells were subjected to *in vitro* MCM loading assay under different conditions: with or without ATP, Cdc6, or ORC. ARS DNA-bound materials were eluted by SDS and analyzed by Western blotting with anti-Mcm3 and anti-Mcm6 antibodies. D to H) GFP-Mcm3 cells (HZY3037) transformed with *pRS315-3xFLAG-mcm3* plasmids were arrested in the G1 phase, and the corresponding whole cell extracts (WCE) were subjected to in vitro MCM loading assay. ARS DNA-bound materials were eluted by LDS and analyzed by Western blotting with anti-Mcm3 and anti-Flag antibodies.
(TIF)

**S7 Fig.** A) AlphaFold predicted interaction between the basic residue at P2 (left panel) or P3 (right panel) in Mcm3SV40-NLS-allR (764KSPRRRRRV772) and Kap60. B) The distances between residues at P2 or P3 within Mcm3's NLS and key residues in Kap60 were modeled by AlphaFold. Orange indicates that, according to AlphaFold 3 modeling, no specific contact was detected that resembled the published structure of SV40-NLS/karyopherin alpha complex.
(TIF)

## Acknowledgements

We thank members of the Zhou lab for helpful discussions and critical readings of the manuscript, Drs. John Diffley (Crick Institute) and Stephen Bell (MIT) for sharing plasmids and yeast strains, Dr. Christopher Putnam (UCSD) for generous assistance with fluorescence microscopy, Dr. Frank Furnari (UCSD) for access to BD LSR II Flow Cytometer, Yujin Lee and members of the Zhou lab for technical assistance.

## Author contributions

**Conceptualization:** Ziyi Wang, Huilin Zhou.

**Data curation:** Ziyi Wang, Raymond T. Suhandynata, Huilin Zhou.

**Formal analysis:** Ziyi Wang, Raymond T. Suhandynata, Huilin Zhou.

**Funding acquisition:** Raymond T. Suhandynata, Huilin Zhou.

**Investigation:** Ziyi Wang, Yun Jing Zhang, Qian-yi Zhang, Kate Bilsborrow, Matthew Leslie.

**Methodology:** Ziyi Wang.

**Project administration:** Huilin Zhou.

**Supervision:** Raymond T. Suhandynata, Huilin Zhou.

**Validation:** Ziyi Wang, Huilin Zhou.

**Writing – original draft:** Ziyi Wang, Huilin Zhou.

**Writing – review & editing:** Ziyi Wang, Raymond T. Suhandynata, Huilin Zhou.

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
