## [Decision Letter · Decision Letter 0]

17 Dec 2024

PGENETICS-D-24-01340

Sequence specificity of an essential nuclear localization sequence in Mcm3

PLOS Genetics

Dear Dr. Zhou,

Thank you for submitting your manuscript to PLOS Genetics. After careful consideration, we feel that it has merit but does not fully meet PLOS Genetics's publication criteria as it currently stands. Therefore, we invite you to submit a revised version of the manuscript that addresses the points raised during the review process.

Please submit your revised manuscript within 30 days Jan 16 2025 11:59PM. If you will need more time than this to complete your revisions, please reply to this message or contact the journal office at plosgenetics@plos.org. Please include the following items when submitting your revised manuscript:

We look forward to receiving your revised manuscript.

Kind regards,

Giovanni Bosco, Ph.D.

Section Editor

PLOS Genetics

Aimée Dudley

Editor-in-Chief

PLOS Genetics

Anne Goriely

Editor-in-Chief

PLOS Genetics

**Additional Editor Comments (if provided):**

Thank you for submitting your manuscript to PLOS Genetics. I am including the reviewer comments below, which I hope you will find useful and constructive. As you will see, they express interest in the study, but they also have several criticisms and suggestions. If it is possible to address the concerns raised with additional data and/or discussion, we would be interested in considering a revised version of the manuscript.

**Journal Requirements:**

**Reviewers' comments:**

Reviewer's Responses to Questions

**Comments to the Authors:**

Reviewer #1: The cellular localization of yeast MCM helicase is cell-cycle controlled, and it localizes in the nucleus and cytoplasm in the G1 and G2-M phases, respectively (Labib et al. NCB, 1999; Niguen et al. Curr Biol., 2000). MCM comprises six subunits, two of which (MCM2 and MCM3) contain a nuclear localization signal (NLS) (Young et al., Gene Cells, 1997; Liku et al. MBC, 2005).

Taking these previous studies, Wang et al. examined the sequence specificity of NLS in yeast MCM3. The authors tested various mutants having an alanine substitution(s) within the MCM3 NLS and investigated the cell viability (Figs. 1 and 2) and the cellular of the mutant proteins (Figs. 3 and 4). The authors revealed that K768 is the most critical, and R769 plays a secondary role. The authors further confirmed these findings by looking at the chromosome association of the MCM3 mutants (Fig. 5). Finally, the authors used AlfaFold3 to investigate the interaction between the MCM3 mutants and Kap60 and discovered that the basic residue at P2 and P2 positions (MCM3 K768 and R769, respectively) interacts with the Kap60 binding pocket in a manner similar to that observed in SV40 NLS.

All experiments were conducted carefully, and the presented results are convincing. I do not find a significant problem in the manuscript. However, the authors should consider whether the mutations, particularly MCM3 K768A, will not affect the formation of the MCM2-7 complex. If the complex formation was compromised, the mutant cells would be inviable, and the mutant protein might stay in the cytoplasm.

Additionally, the authors should be able to show the location of MCM3 NLS within the MCM2-7 complex (related to Fig. 1A). I am asking this because previous publications suggested that MCM2-7 is likely imported and exported to the nucleus as the MCM 2-7 complex (e.g. Liku et al. MBC, 2005).

Minor point

Fig. 3B and Fig. 4B: The labelling for "nuclear" and "cytosolic" is opposite.

Reviewer #2: This manuscript provides a comprehensive analysis of the NLS of Mcm3, including analysis of cell viability, nuclear localisation of GFP-Mcm3 using fluorescence microscopy, biochemical assays to assess capacity of MCMs to load in vitro, and AF3 predictions. Overall the manuscript is well written, the results are clear and convincing and make sense. Given the often ambiguous nature of NLSs, this study provides new insight into what constitutes a monopartite NLS, and suggest that AF3 predictions could serve as a useful tool to guide future studies of NLS function by assessing NLS-importin contacts of different mutants. Overall I think it is suitable for publication, but several minor issues need to be addressed, as detailed below:

Line 91: “structure-guided approach”. The AF3 predictions are in the very end of the paper as the last figure and were not really used for a structure-guided approach. As they are presented, the AF3 predictions provide a potential structural basis for the observed biochemical and genetic results. However, the amount of novel information gained from these models is of limited value, especially given the existing crystal structure of the SV40-NLS with importin alpha. There are no predictions that were generated and tested based on the AF3 data. I would rephrase to say the the AF3 predictions could serve as a way to study NLS function, but that this requires testing on other NLSs in future studies, and would also point out the limitations.

Line 104: Perhaps it is worth explaining the plasmid shuffle concept – the way it’s presented is not clear to those not familiar with classic yeast genetic approaches. E.g. – “Mcm3 is an essential gene and cannot be deleted or mutated. To study mcm3 mutants, we employed a genetic approach that allows us to conditionally remove WT Mcm3 (by selecting against a URA3 plasmid with 5-FOA) while simultaneously complementing with either WT or mutant forms of Mcm3 (selected for with the LEU2 marker).”

Line 139: I would add here that this result also fits well with the fact that K768 is highly conserved and is the only invariant K residue across evolution.

Line 188: I would add here – “we considerd that mutations introduced into Mcm3 could directly affect MCM loading, leading to the observed defect in cells. To test this, we determined the capacity of these mutants to be loaded on DNA in vitro”. I would add at the end of this paragraph “Therefore, the observed reduction in MCM loading in cells is not because mutants cannot be loaded, but rather because they are not imported into the nucleus”.

Figure 1A: R769 and R771 are also highly conserved and are basic residues in mice/human. These should also be highlighted. However, the only invariant residue is K768.

Figure 2C: the annotation of the mutations in the table on the right is incorrect for the two bottom mutants. These should be SV40-NLS and SV40-NLS-allR. The cell growth annotation is also incorrect (should be ++ for SV40-NLS). This table was clearly copied from Figure 2A but not completely edited to reflect the result in 2C…

Figure 4B: The colors of the graph and the legend seem to have been swapped. The dark part in the graph is clearly nuclear, not cytosolic.

**Have all data underlying the figures and results presented in the manuscript been provided?**

Reviewer #1: Yes

Reviewer #2: Yes

PLOS authors have the option to publish the peer review history of their article (what does this mean? ). If published, this will include your full peer review and any attached files.

**Do you want your identity to be public for this peer review?** For information about this choice, including consent withdrawal, please see our Privacy Policy .

Reviewer #1: No

Reviewer #2: **Yes:** Gideon Coster

**Figure resubmission:**

While revising your submission, please upload your figure files to the Preflight Analysis and Conversion Engine (PACE) digital diagnostic tool, https://pacev2.apexcovantage.com/ . PACE helps ensure that figures meet PLOS requirements. To use PACE, you must first register as a user. Registration is free. Then, login and navigate to the UPLOAD tab, where you will find detailed instructions on how to use the tool. If you encounter any issues or have any questions when using PACE, please email PLOS at figures@plos.org. Please note that Supporting Information files do not need this step. If there are other versions of figure files still present in your submission file inventory at resubmission, please replace them with the PACE-processed versions.
---

## [Editor Report · Decision Letter 1]

2 Jan 2025

Dear Dr Zhou,

We are pleased to inform you that your manuscript entitled "Sequence specificity of an essential nuclear localization sequence in Mcm3" has been editorially accepted for publication in PLOS Genetics. Congratulations!

Yours sincerely,

Katsumi Kitagawa, PhD

Guest Editor

PLOS Genetics

Giovanni Bosco

Section Editor

PLOS Genetics

Aimée Dudley

Editor-in-Chief

PLOS Genetics

Anne Goriely

Editor-in-Chief

PLOS Genetics

Comments from the reviewers (if applicable):

**Data Deposition**

http://datadryad.org/submit?journalID=pgenetics&manu=PGENETICS-D-24-01340R1

More information about depositing data in Dryad is available at http://www.datadryad.org/depositing . If you experience any difficulties in submitting your data, please contact help@datadryad.org for support.

**Press Queries**
